# Synergistic Effects of Probiotics and Phytobiotics on the Intestinal Microbiota in Young Broiler Chicken

**DOI:** 10.3390/microorganisms7120684

**Published:** 2019-12-11

**Authors:** Hao Ren, Wilfried Vahjen, Temesgen Dadi, Eva-Maria Saliu, Farshad Goodarzi Boroojeni, Jürgen Zentek

**Affiliations:** Institute of Animal Nutrition, Freie Universität Berlin, Königin-Luise-Str. 49, 14195 Berlin, Germany; Wilfried.Vahjen@fu-berlin.de (W.V.); temehi@gmail.com (T.D.); Eva-Maria.Saliu@fu-berlin.de (E.-M.S.); farshad.goodarzi@fu-berlin.de (F.G.B.)

**Keywords:** feed additives, probiotics, phytobiotics, symbiotics, gut microbiota, antibiotic-resistant *Enterobacteriaceae*

## Abstract

Probiotics and phytobiotics have been studied as in-feed antibiotic alternatives for decades, yet there are no studies on their possible symbiotic effects. In the present study, newly hatched chickens were fed with feeds supplemented either with host-specific *Lactobacillus* strains (*L. agilis* and *L. salivarius*), commercial phytobiotics, or combinations of both. After 13 days of life, crops and caecums were analyzed for bacterial composition (16S rDNA sequencing, qPCR) and activity (bacterial metabolites). Crop and caecum samples were also used to study the ex vivo survival of a broiler-derived extended-spectrum beta-lactamase (ESBL) producing *Escherichia coli* strain. In the crop, combinations of probiotics and phytobiotics, but not their single application, increased the dominance of lactobacilli. The single application of phytobiotics reduced the metabolite concentrations in the crop, but certain combinations synergistically upregulated the metabolites. Changes in the qualitative and quantitative composition of the caecal microbiota were less pronounced than in the crop. Acetate concentrations were significantly lower for phytobiotics or the *L. agilis* probiotic strain compared to the control group, but the *L. salivarius* probiotic showed significantly higher acetate concentrations alone or in combination with one phytobiotic. The synergistic effects on the reduction of the ex vivo survival of an ESBL producing *E. coli* strain in crop or caecum contents were also observed for most combinations. This study shows the beneficial synergistic effects of probiotics and phytobiotics on the intestinal bacterial composition and their metabolic activity in young broilers. The reduced survival of potentially problematic bacteria, such as ESBL-producing *E. coli* further indicates that combinations of probiotics and phytobiotics may lead to a more enhanced functionality than their individual supplementation.

## 1. Introduction

Poultry meat production is expanding rapidly, as global meat consumption is constantly increasing [1]. As antibiotic growth promoters (AGPs) are still used in many countries to increase performance in poultry production, their usage increased simultaneously. However, AGPs contribute to the development and increase of antibiotic-resistant bacteria such as extended-spectrum beta-lactamase (ESBL) producing bacteria in poultry [2]. Many countries have, therefore, banned AGPs for environmental and public health concerns [3,4]. AGPs in poultry production are well known to increase animal performance and infection resistance, and thus, alternatives are demanded to compensate for this loss [5,6]. Among the commercially existing alternatives, probiotics and phytobiotics have been reported to impact on the performance and health in poultry [7].

Regarding probiotics, many publications have shown health promoting effects that are associated with the modifications on gut microbiota [8,9]. *Bacillus* spp., yeasts, and lactic acid bacteria are commonly used as probiotics in animal nutrition. Among the lactic acid bacteria, the lactobacilli have the advantage that they are “generally regarded as safe” (GRAS-status) and are indigenous to the intestinal tract. Certain *Lactobacillus agilis* strains are able to modify the presence of pathogenic bacteria in vitro [10] and ex vivo [11] or regulate the gut microbiota in broiler chickens in vivo [12,13]. In some studies, several *L. salivarius* strains have also been shown to promote animal health [14,15]. Proper supplementation of certain probiotic strains may also lead to the immunomodulation of the host [16], and consequently, resilience against pathogenic bacteria may be increased [17,18]. However, those effects primarily depend on the specific strain, and it is still unclear if immunomodulation is a direct effect of the probiotics or a response to a modified microbiota. Nevertheless, the health-promoting effect of probiotics has often been observed, and this effect may sometimes lead to an improvement in performance. Finally, host-specific probiotics were reported to show better survival and colonization of the strain in the intestinal tract of the host [19,20].

Phytobiotics have also been studied as an alternative to AGPs due to their strong antibacterial activity in vitro and their beneficial influence on animal health and performance in broiler production systems [21,22,23]. Phytobiotics have the potential to inhibit pathogenic bacteria [24] and show a range of host-related responses such as improvement in antioxidative status [25], intestinal barrier functions [26] as well as a beneficial modification the digestive or immune function of the host [27]. These modes of action may contribute to an increase in growth performance [28,29]. However, like probiotics, due to a lack of studies, it is still unclear if these effects are directly induced by the presence of the phytobiotics or are an indirect effect of changes within the intestinal microbiota.

Synergistic feed additives are thought to act by combining their individual effects that lead to a superior effect than their sole application [30]. This principle has been applied for combinations of probiotics and prebiotics with the intent to promote beneficial bacteria and at the same time, supply suitable substrates for the probiotic [31]. There are also reports on other combinations of feed additives like organic acids combined with phytases [32], probiotics [33], or phytobiotics [34]. To our best knowledge, only one study evaluated the use of a probiotic *E. faecium* strain with a commercial phytobiotic product as a combination in broiler chickens. This study showed beneficial effects on animal health, but no effect on animal performance [35]. In summary, the use of such possibly synergistic combinations of feed additives has not received attention in the field of poultry nutrition. Therefore, a concept was designed and applied in this study to combine host-specific probiotic *Lactobacillus* strains with specific phytobiotics to invoke beneficial synergistic effects for the animal. 

Both probiotics and phytobiotics are known to modulate the intestinal microbiota in poultry [36]. Therefore, studies on possible synergistic effects must include an in-depth analysis of the bacterial composition and activity. Furthermore, in light of the evolution of ESBL-producing *Enterobacteriaceae* in poultry [2], the impact of the feed additives on the colonization of antibiotic-resistant bacteria is of high interest.

As young animals are still in the process of developing a mature microbiota [37], this progress may be modified more easily via feed additives [9]. Thus, young broiler chickens were used to investigate the effect of these feed additives on the intestinal microbiota.

Thus, the aim of this study was to compare two different *Lactobacillus* strains and two different commercial phytobiotics as well as their combinations on the possible synbiotic activity in young broiler chickens. 

## 2. Materials and Methods

### 2.1. Ethics Statement

The feeding trial was conducted according to the German Animal Welfare Act (TierSchG) and approved by the local state office of occupational health and technical safety ‘Landesamt für Gesundheit und Soziales, Berlin’ (LaGeSo Reg. A 0437/17).

### 2.2. Animals, Rearing Conditions, and Experimental Diets

Newly hatched Cobb 500 broiler chicks were purchased from a commercial hatchery and randomly allocated into nine experimental groups. All animals were kept in cages and had ad libitum access to feed and water. The ambient temperature was adjusted as follows: for the first 2 days of age, the ambient temperature was 34 °C and was then gradually reduced by 3 °C per week. Artificial light was provided continuously during the first 3 d of age. From 4 d of age onwards, the lighting regime consisted of an 18 h light and 6 h dark cycle. Each group included 21 animals in seven replicate cages per group. Nine experimental feeds were offered in meal form as follows: control feed, two probiotic feeds (10^10^ cfu/kg feed), two phytobiotic feeds (250 mg/ kg feed), and four feeds with the respective combination of probiotics and phytobiotics at the same concentrations. Feed composition and nutrient content of the basal diet are shown in Table 1.

### 2.3. Sampling

On the 13th day of life, two birds from each group were randomly selected daily, weighed, and sacrificed by exsanguination after anesthesia. In order to obtain fresh digesta samples for ex vivo studies and their appropriate processing, this procedure was continued until 10 animals per group were sampled in a span of five days. The digesta from crop and caecum of birds were collected individually and allocated into two portions. One portion was snap-frozen in liquid nitrogen for DNA extraction and metabolite analysis; the other portion was used for ex vivo studies with an ESBL-producing *E. coli* strain.

### 2.4. Bacteria and Media

Probiotic LS1 (*L. salivarius*) and LA73 (*L. agilis*) were isolated, identified, and characterized previously [38]. Both strains were cultured anaerobically (starting culture) or aerobically (biomass production) in de Man, Rogosa, and Sharpe broth (MRS, Carl Roth GmbH + Co. KG, Germany) at 37 °C for 24 h.

The ESBL-producing *E. coli* (EE10716) strain was isolated from the broiler chicken by the Institute of Microbiology and Epizootics of the Freie Universität Berlin within the RESET project (Germany, http://www.reset-verbund.de/index.htm) and produced the CTX-M-15 lactamase [39]. This strain was selected as the model strain to study its survival in ex vivo broiler ingesta samples. The strain was stored as cryo stock and cultured in brain heart infusion broth (BHI, Carl Roth GmbH + Co. KG, Karlsruhe, Germany) for further application.

### 2.5. Feed Additives

The probiotic *Lactobacillus* cells were harvested from MRS broth after aerobic growth at 37 °C for 12 h. The pelleted biomass was concentrated by centrifugation (10 min, 15000× *g*, 4 °C) and freeze-dried at –55 °C in a lyophilizer (LyoVac GT2, Hürth, Germany) for 48 h after pre-freezing at –80 °C as described previously [40]. The lyophilized probiotics were stored at 4 °C until mixed with the feed. 

The commercial phytogenic compounds (formulation C and L) were provided in solid form by EW nutrition (Visbek, Niedersachsen, Germany) and kept at 4 °C until mixed with feed. The active ingredients in formulation C were carvacrol and cinnamaldehyde, while formulation L contained carvacrol, cinnamaldehyde, and eugenol. All additives were mixed with the basal diet using a feed mixer and stored at room temperature. To ensure that sufficient amounts of viable probiotic cells and concentrations of the volatile phytobiotic products were present in the feeds, the diets were prepared on a weekly basis.

### 2.6. 16S rDNA Sequencing and qPCR

Total DNA was extracted from 0.2 g digesta samples of crop and caecum (45 samples for each section, five samples for each group) with a commercial extraction kit (QIAamp Fast DNA stool mini kit, Qiagen, Hilden, Germany) in accordance with the manufacturer’s instructions with minor modification (lysis step at 90 °C). The resulting DNA extracts were stored at –30 °C until further analysis.

DNA extracts were subjected to amplicon sequencing using an Illumina NextSeq500 sequencer (LGC, Berlin, Germany) with two 150–base pair reads. After a combination of forward and reverse reads using the BBMerge tool [41] and demultiplexing, the resulting 16S-rDNA sequences were analyzed using the QIIME2 pipeline [42] and the SILVA SSU database [43]. Quality control and determination of sequence counts were performed using the DADA2 [44]. Sequence variants with less than five counts were excluded from further analysis to increase the confidence of sequence reads and reduce bias by possible sequencing errors [45]. Normalization of sequence reads was done by verification with an equal representation of 10,000 sequences per sample [46]. Further statistical analysis was done using RStudio (Rstudio, Boston, USA) and KNIME 4.01 (KNIME, Zürich, Switzerland) [47] with the R package stats and ggplot2.

Quantification of predominant bacterial groups and species in poultry was performed via qPCR assays with five biological replicates per group (Appendix A). Target gene copy numbers were calculated from standard curves with known copy number concentrations.

### 2.7. Analysis of Bacterial Metabolites

Short-chain fatty acids (SCFA) in crop and caecum contents were analyzed as described previously [48]. In short, SCFAs were analyzed using gas chromatography (Agilent Technologies 6890N, autosampler G2614A, and injection tower G2613A; Network GC Systems, Böblingen, Germany) equipped with a flame ionization detector. D- and L-lactate was measured by HPLC (Agilent 1100; Agilent Technologies, Böblingen, Germany) with a pre-column (Phenomenex C18 4.0 4.0 × 2.0 mm; Phenomenex Ltd., Aschaffenburg, Germany) and an analytical column (Phenomenex Chirex 3126 (D)-penicillamine 150 × 4.6 mm; Phenomenex Ltd.).

### 2.8. Ex-vivo Survival of the ESBL-Producing E. coli Model Strain in Crop and Caecum

The ex vivo survival of the ESBL-producing *E. coli* model strain was evaluated according to a previous study with minor modification [49] in a two-step incubation assay. In short: in the first step, fresh crop and caecum samples were immediately diluted anaerobically in incubation buffer at 1:5 (*w/v*). After 5 min sedimentation, supernatants (190 µL) were anaerobically transferred to microtiter plates, and the *E. coli* strain EE10716 was added (a final concentration of 2 × 10^5^ cells/mL). To avoid the interference from the indigenous ESBL-producing bacteria in the samples, identical plates but without inoculation of the ESBL indicator strain was prepared and incubated under the same conditions. In the second step, after anaerobic incubation at 37 °C for 12 h, samples were transferred to a microtiter plate containing Mueller–Hinton broth 2 (Carl Roth GmbH + Co. KG, Germany) with 4 µg/ml cefotaxime to ensure selective growth of the resistant *E. coli* strain. The plates were then incubated aerobically in a microtiter plate reader (Tecan infinite M Plex, Männerdorf, Switzerland) at 37 °C and OD_690nm_ was obtained as the OD_experiment_ subtracted by the OD_control_. The resulting growth curves were subjected to a non-linear regression model using the Gompertz equation, which gave the best fit with an r^2^ ≥ 0.98 for all samples. The lag time was then documented as an indicator for *E. coli* growth inhibition. All assays were performed with five technical replicates. 

### 2.9. Statistical Analysis

Results are presented as means and pooled standard error of mean except for figure 2, which is presented as the means and standard deviation. Due to the non-normally distributed nature of the data, we chose to use the Kruskal–Wallis test, followed by the Mann–Whitney test, when appropriate. The Chi-square test was performed to compare *Clostridium. perfringens* (*C. perfringens*) counts. Statistical procedures were performed at a significance level of 95% using the SPSS Statistics software (SPSS, Chicago, USA).

## 3. Results

### 3.1. Qualitative Determination of the Intestinal Microbiota in Young Broiler Chickens

A total of 1.26 × 10^6^ quality sequence reads from 89 samples (44 crop samples and 45 ceca samples) with a mean combined read length of 404 nucleotides were used for the qualitative analysis of the bacterial composition. One sample (control group, crop) had to be omitted from the analysis, as it contained a massive amount of *Aeromonas* spp., and *C. perfringens*, which classified this sample as a true outlier.

A total of six phyla, 18 orders, and 88 genera were assigned to the sequences. A comprehensive overview of the taxonomic assignments is given in Appendix A.

Overall, the crop was heavily dominated by *Lactobacillus* spp., while caecal contents displayed a more even distribution with an unidentified *Clostridiales*, *Bacteroides* spp., and *Faecalibacterium* spp. as the most dominant genera (Figure 1).

### 3.2. Impact of Probiotics and Phytobiotics on the Relative Composition of the Crop Microbiota

The crop was considerably dominated by *Lactobacillus* spp. sequences in all feeding groups, ranging from 98.0% to 99.7% abundance (Appendix A). Other dominating genera, including *Aeromonas* spp., *Acinetobacter* spp., *Bacteroides* spp., as well as two unidentified genera belonging to the *Clostridiales* order rarely exceeded one percent of all sequences. The control group, as well as treatment groups with single feed additives, showed the lowest lactobacilli abundance. However, combinations of probiotics and phytobiotic additives significantly increased lactobacilli abundance at the expense of the next dominant genera (*Aeromonas*, *Acinetobacter, Bacteroides*) compared to control. One exception was the combination of LA73 and formulation C, in which the *Clostridium sensu stricto 1* genus and *Bacteroides* spp. prevailed. Seven different *Lactobacillus* species were assigned based on their unique OTUs, which significantly differed in their abundance in the crop (Table 2). As expected, supplementation of the probiotic *Lactobacillus* strains significantly increased their abundance in the respective treatment groups. Another dominating *Lactobacillus* species (*L. crispatus*) showed a significant numerical reduction in all treatment groups compared to the control.

Interestingly, the putative pathogenic genera *Aeromonas* spp. and *Acinetobacter* spp. were among the dominating sequences in the crop of 13-day-old animals. No significant differences were observed in this genus, but combinations of probiotics and phytobiotic additives strongly reduced the numerical abundance of *Aeromonas* spp., except for the combination of LS1 and formulation C. The genus *Acinetobacter* spp. was strongly reduced in the LA73 and formulation C groups, as well as in all combination groups (see Appendix A).

Ecological indices showed no significant differences for richness (number of different sequences), but diversity (Shannon Index) and accordingly evenness were significantly reduced in treatment groups supplemented with the phytobiotic formulation C as well as the combination of the probiotic LS1 and formulation L (Appendix A).

### 3.3. Impact of Probiotics and Phytobiotics on the Relative Composition of the Caecal Microbiota

An unidentified *Clostridiales* genus and *Bacteroides* spp. were the most dominant genera in the caecum, followed by *Faecalibacterium* spp. (Appendix A). Due to high individual differences in the samples, only a few significant differences were observed. There was a trend for an increased abundance of the unidentified *Clostridiales* genus in the probiotic LS1 and in the LA73/ formulation L treatment groups, which was offset by a significant reduction of *Faecalibacterium* spp. in these groups. Additionally, compared to control- or phytobiotic supplemented feed groups, *Anaerostipes* spp. showed significant differences in feed groups that were supplemented with the probiotic strain LS1 alone or in combination.

Ecological indices did not differ significantly between treatment groups (Appendix A). A range of 40.6 to 45.8 independent OTU were found with even distributions of diversity and very similar evenness.

### 3.4. Quantitative Determination of the Intestinal Microbiota in Young Broiler Chickens

16S rDNA sequencing yields an in-depth view of the abundance of bacterial genera but is unable to quantify the bacterial composition in the intestinal tract. Therefore, qPCR assays on a range of dominant bacterial groups and species were carried out.

### 3.5. Impact of Probiotics and Phytobiotics on Dominant Bacterial Groups and Species in the Crop

The quantitative determination of the crop microbiota confirmed the dominance of lactobacilli and mirrored results for relative *Lactobacillus* spp. abundance (Table 3). The most prominent species was *L. salivarius*, followed by *L. reuteri*, *L. agilis* and *L. acidophilus*. As expected, supplementation of LS1 significantly increased the *L. salivarius* counts, while LA73 significantly increased *L. agilis* copy numbers. This was also visible for their combinations with the phytobiotic additives. Interestingly, the supplementation of the probiotic *L. salivarius* strain reduced the counts of the *L. agilis* species and vice versa. The other dominant lactobacilli (*L. reuteri* and *L. acidophilus*) showed no significant response to the presence of either probiotic supplementation.

The combination of LS1 with both formulation C and formulation L significantly reduced bacteria belonging to the Clostridial Cluster XIVa, while no changes were observed in any other feeding group. The copy numbers of the total enterobacteria and the *Escherichia* group were higher in the LA73 group, but absolute differences were only minor. However, in combination with either phytobiotic additive, a small (formulation C) or drastic (formulation L) reduction was observed for *Escherichia* group counts. Similarly, the copy numbers of the integrase 1 gene, responsible for the incorporation of foreign DNA, mirrored the trend seen for the *Escherichia* group. 

Finally, due to the presence of an unidentified *Clostridium sensu stricto* 1 in the sequencing data, we also tested for the presence of *C. perfringens*, which is phylogenetically closely related to this genus (Appendix A). *C. perfringens* was detected sporadically in the crop samples, but no significant differences were observed (*p* = 0.103, Chi-square test). However, all the samples from the feed groups supplemented with LA73 or its combination with formulation L were negative for *C. perfringens*. Interestingly, the omitted sample from the control group showed a very high amount of *C. perfringens* (log 6.1).

### 3.6. Impact of Probiotics and Phytobiotics on Dominant Bacterial Groups and Species in the Caecum

The supplementation of probiotics and phytobiotics showed no significant impact on the caecal microbiota (Table 4). Two clostridial clusters, the *Bacteroides-Prevotella-Porphyromonas* cluster, and enterobacteria followed by lactobacilli dominated the caecum of 13-d old broiler chickens. *C. perfringens* was detected in all control samples (see Appendix A), but only rarely in the other experimental groups. However, no significant differences between the feed groups were observed (*p* = 0.126, Chi-square test).

### 3.7. Bacterial Metabolism of the Intestinal Microbiota in Young Broiler Chickens

Short-chain fatty acids and lactate concentration in the crop are shown in Table 5. As expected, the crop was dominated by lactate in all feeding groups, while acetate and propionate only played minor roles in the formation of bacterial metabolites. The phytobiotic supplementation significantly reduced lactate concentrations in the crop, and also showed the numerically lowest concentrations of acetate. Surprisingly, the single supplementation of the probiotic lactobacilli did not increase lactate concentrations compared to the control. However, the combination of LA73 and formulation L led to significantly more lactate than in any other treatment group. The combination of the probiotic LS1 with phytobiotic products also led to higher lactate concentrations than their single addition.

Lactate in the caecum is not shown, as those values are generally very low or undetectable. The dominating SCFA in the caecum was acetate (Table 6). The phytobiotic products again showed significantly (formulation C) or numerically but non-significantly (formulation L) lower acetate concentrations compared to the control group. The addition of the probiotic strain led to a diverse bacterial response in the caecum, as the strain LS1 showed a significantly increased acetate concentration, while L73 led to significantly lower acetate concentrations compared to the control group. However, all combination groups displayed significantly or numerically higher concentrations than their respective single addition except for the combination of LS1 and formulation C.

### 3.8. Ex-vivo Growth Response of an ESBL Producing E. coli Model Strain in Intestinal Samples

The impact of different intestinal conditions due to the addition of probiotics and phytobiotics was tested for the ex vivo survival of an ESBL-producing *E. coli*. Lag time, i.e., start of the exponential growth phase, is the most informative fitness parameter of a bacterium, and thus, this parameter was used to estimate the impact of probiotics and phytobiotic supplementation.

In crop samples, the significantly lowest lag time, and thus, the best fitness of the *E. coli* strain was observed in the LS1 treatment group, followed by the formulation L (Figure 2A). However, the combination of both additives slightly increased lag time compared to a single application. The significantly lowest fitness was noted in the combination of LA73 and formulation L, although their single application yielded a significantly higher fitness. 

In caecal samples, the best survival of the *E. coli* strain was observed in the control group (Figure 2B). Both additive types showed a numerically lower fitness as single supplementation, but combinations of probiotics and phytobiotics displayed a significantly lower survival with the exception of the combination of LS1 and formulation C.

## 4. Discussion

Accumulating numbers of studies show that novel additives such as probiotics or phytobiotics may be used as alternatives to in-feed AGPs. However, the efficiency of the alternative additives depends on many factors like uptake concentration, overall diet, supplementation method, or the rearing environment [50]. To maximize the efficiency of those alternatives, the combination in accordance with a synergistic concept is a favorable solution that may act beyond their single applications. The present study investigated the synergistic effects of probiotics and phytobiotic feed additives on the intestinal microbiota in young broiler chickens. The gut microbiome is a key to understand animal health and nutrition better [51], and thus, this study focused on the bacterial composition and –activity in crop and caecum of young broiler chickens that has not yet developed a stable microbiota.

Probiotics generally do not reduce the total amount or activity of bacteria in the gut, but can sometimes increase bacterial metabolite concentrations in broiler chicken [52,53]. On the other hand, phytobiotics are often used due to their strength in vitro antibacterial activity [54]. The active ingredients in the phytobiotic products were carvacrol and cinnamaldehyde as well as additionally eugenol in formulation L. All three substances have been shown to inhibit a range of bacteria in vitro and show diverse effects on performance, immunology, and reduction of pathogenic bacteria in broiler chicken [55,56,57]. The probiotic *Lactobacillus* strains in this study were previously characterized in vitro regarding their resilience against both phytobiotic products (data not shown). Both strains showed a high tolerance in vitro, which could be confirmed in vivo, especially for the *L. salivarius* strain. In fact, a strong synergistic effect was observed for the species *L. salivarius* with formulation L, which may indicate that the functionality of LS1 increased accordingly when applied as a synergistic product. However, the species *L. agilis* was strongly inhibited by the effects of the formulation C in vivo, but in combination with LA73 an increase of this species was observed in the crop. This also points to a synergistic effect for increased LA73 colonization in combination with formulation C. Taken together; the data suggest that indigenous *L. agilis* strains may be much more sensitive to phytobiotic pressure compared to the supplemented *L. agilis* strain. This reflects strain-specific differences in lactobacilli in general. Therefore, synergistic effects seem to be in effect regarding certain combinations of probiotic and phytobiotics. Unfortunately, to the best knowledge of the authors, there are not many reports on phytobiotic modifications of the intestinal microbiota in broiler chicken, studies on combined usage with probiotics are even rare [35]. However, studies in humans, pigs, and rats show that the absorption of the mentioned essential oils occurs in the upper small intestine [58,59]. It is therefore probable that pancreatic enzymes in poultry attack these substances, and resorption of their metabolites could be expected before they reach the caecum. Consequently, it is unlikely that relevant concentrations of the phytobiotics reached the hindgut. This implies that results on crop and caecum microbiota should be viewed separately from different angles and that modifications of the caecal microbiota are largely due to bacterial- or host-related changes in the upper intestinal tract.

The crop plays an essential role in the transient storage and moisturization of feed [60]. It is also viewed as a pre-gastric fermentation chamber that defines the input of bacteria into the gut [61]. Generally, the crop of broiler chicken is heavily dominated by certain dominant *Lactobacillus* species [62,63,64], which was also observed in this study. Of the few studies on the subject, one report with a probiotic *L. salivarius* strain showed no effect on crop lactobacilli after administration [65]. This was also observed for the number of lactobacilli in this study, but the single addition of probiotic strains significantly enhanced their quantity compared to the control group. Thus, both strains were able to colonize the crop. This was not unexpected, as both probiotic strains were originally isolated from the broiler intestine and already demonstrated great potential for in vitro survival under-stimulated gastric stress and epithelial adherence in our previous study [38].

Significant positive synergistic effects on relative *Lactobacillus* spp. abundance were only observed for *L. salivarius* in combination with formulation L as well as for *L. agilis* with both phytobiotics. In general, the supplementation of the probiotic strains seemed to be the overriding effect on *Lactobacillus* spp. abundance, while the additional phytobiotic supplementation showed only minor effects. Similarly, a slight non-significant decrease of species richness was observed for combination groups, but significant differences for microbiota diversity (Shannon-index) did not show clear synergistic effects. 

The impact of the probiotics and phytobiotics on the crop microbiota also extended to non-dominant bacteria. For instance, the *Clostridium sensu stricto* 1 genus exhibited the highest abundance apart from the lactobacilli. This *Clostridium* genus has been shown to be associated with necrotic enteritis and *Clostridium perfringens* infection models [66,67]. However, the *Clostridium sensu stricto* 1 cluster also contains species such as *C. butyricum*, which has also been used as a probiotic in poultry [68]. It is, therefore, difficult to assign a positive, indifferent or negative role to this genus. Nevertheless, the comparison of *Clostridium sensu stricto* 1 sequencing data to the much more sensitive *C. perfringens* qPCR data did not show any correlation (data not shown). We can, therefore, conclude that this genus probably did not include *C. perfringens*. The abundance of *Clostridium sensu stricto* 1 was high in single probiotic and formulation L supplemented feed groups but was dramatically reduced in combinations of LS1 with both phytobiotic products and especially in LA73 with formulation L. Thus, synergistic effects in the significant reduction of this *Clostridium* spp. were visible only for certain combinations. Although the additional eugenol in formulation L may have played a role in enhancing the *Clostridium sensu stricto* 1 abundance compared to formulation C in single supplementation; this does not account for its total inhibition in combination with both probiotic strains. These results signify again that the synergistic mode of action on certain bacteria are not additive but rely on the impact of the feed additives on other bacteria. In this case, the concomitant responses of *Faecalibacterium* spp., *Blautia* spp., and an unidentified *Clostridiales* may have played a role in the significant modification of *Clostridium sensu stricto* 1.

Interestingly, similar changes in relative abundance were observed for the putatively pathogenic genera *Aeromonas* spp. and *Acinetobacter* spp. Furthermore, *C. perfringens* positive samples in the crop were generally lower in combination groups, which points to their potential to a synergistic potential to reduce detrimental bacteria in the intestinal tract. Nevertheless, synergistic effects for these bacteria were not visible for all combinations. Therefore, the response of the intestinal microbiota to different phytobiotics seems to be quite diverse. However, as beneficial synergistic effects are clearly visible regarding putatively pathogenic bacteria, the combination of certain probiotic and phytobiotic products may be advantageous for animal health.

Overall, bacterial activity in terms of bacterial metabolites was lower in the crop of feed groups with single phytobiotic addition, although only the reduction in lactate was significant. Carvacrol, cinnamaldehyde, and eugenol are all known to inhibit bacterial growth in vitro, and consequently, their activity [69,70]. Our results indicate that both phytobiotic formulations indeed inhibited bacterial metabolism, although no significant changes in the absolute bacterial counts were observed. Consequently, at the employed in-feed concentrations, the phytogenic products may not inhibit total bacterial growth *per se*, but significantly reduce their activity in the crop. However, the production of lactate or acetate in the intestinal tract is usually considered beneficial, as it may inhibit pathogenic or other bacteria detrimental to the host [71,72]. The increased lactate concentration in certain combinations of phyto- and probiotics points to a beneficial synergistic effect. Still, the synergism seems to depend on specific combinations and cannot be classified as an additive effect of individual supplementation. 

We also monitored the ex vivo survival of an ESBL producing, but non-pathogenic *E. coli* strain in the intestinal contents because ESBL producing enterobacteria have become a worldwide concern in poultry production [73]. The results of our study show synergistic effects on reducing the ex vivo survival of the *E. coli* model strain in crop contents compared to the control. The in vivo results on the quantification of the *Escherichia* group show a similar reducing synergism except for LA73/formulation C. Lactobacilli are known for their antagonistic activity against enterobacteria [74,75] and both probiotic strains showed exceptional inhibitory activity against the *E. coli* strain in vitro and ex vivo [38]. Contrary to these results, data from both the ex vivo assay and the *Escherichia* group quantification showed that the single addition of the probiotic strains had only slight effects on *E. coli* survival. Thus, the inhibitory activity of the phytobiotic products was probably necessary to enhance the impact of the probiotic strains. However, the survival of the *E. coli* strain was also affected in the caecum, where active phytobiotic concentrations are considered low. This indicates that different modes of action may be in play. 

In the context of ESBL producing enterobacteria and transfer of their resistance genes, the presence of the enterobacterial class 1 integron integrase 1 gene (*int1*) was also monitored. This enzyme is a key protein in the incorporation of foreign DNA in enterobacteria [76]. Its copy numbers correlated highly to the count of the *Escherichia* group (*p* < 0.0001; 0.551 coefficient) as well as to the count of enterobacteria (*p* < 0.0001; 0.423 coefficient). However, only the combination of LA73 and formulation L showed a reducing effect on *int*1 concentration. The *int1* gene is widely distributed in enterobacteria, and it is likely that certain enterobacterial species or strains responded differently to the supplementation of the feed additives. Nevertheless, the results clearly show that synergistic effects of probiotics and phytobiotics may be superior to a single addition to combat the spread of enterobacterial antibiotic resistance. 

In the caecum, fermentation of undigested nutrients occurs [77], and the bacterial composition and activity is largely determined by incoming nutrients as well as bacteria from the small intestine [78]. Their metabolites (SCFA) can be used as an energy source by the host and may contribute to meet the energy requirements of the animal. Furthermore, the caecum also determines the output of the potentially detrimental bacteria into the environment and thus has an important impact on stable hygiene.

In this study, the impact of the feed additives on the caecal bacterial composition was much less pronounced compared to changes observed in the crop. This may point to the fact that the active compounds in the phytobiotic products (carvacrol, cinnamaldehyde, and eugenol) are metabolized in the small intestine. Consequently, their active concentration may be drastically reduced. Nevertheless, an antagonistic relationship between the abundance of a dominating unidentified *Clostridiales* genus and *Faecalibacterium* spp. was observed, as the significant reduction of *Faecalibacterium* spp. was always offset by a trend for an increase of the unidentified *Clostridiales* genus. As noticed for other parameters, a trend for synergistic effects was again visible for the combination LA73 and formulation L. 

The dominating bacteria in the caecum are most likely to be the most prominent producers of SCFA from undigested nutrients, and indeed, an increase in the relative abundance of the unidentified *Clostridiales* genus always corresponded with increased acetate concentrations. The increased metabolite production points to an enhanced capacity to ferment undigested nutrients and indicates a more mature microbiota. As a mature microbiota is viewed as beneficial [79], the caecal microbiota, especially in animals, fed the combination LA73 and formulation L, could have developed faster than the microbiota in other feed groups. 

There were two noteworthy exceptions to the generally low response of the caecal microbiota: a reduced colonization of *C. perfringens* and reduced ex vivo survival of the ESBL-producing *E. coli* in all treatment groups. Apparently, adverse conditions for these two detrimental species existed due to the addition of the feed additives. However, as the phytobiotic concentration is believed to be low in the caecum, these adverse conditions may have mainly originated from interbacterial competition or host-related responses that were induced in the crop or small intestine. 

## 5. Conclusions

This study has shown that probiotics and phytobiotics can have beneficial synergistic effects on the intestinal microbiota in young chickens. The impact of the probiotics and phytobiotics was mainly confined to the crop, but synergistic effects were also observed in the caecum regarding the colonization of *C. perfringens* and the survival of an ESBL producing *E. coli* strain. Comprehensively considering the effects in microbiota shifts, changes in bacterial metabolites, and resilience to detrimental bacteria in host GIT, the combination of the *L. agilis* strain in combination with the formulation L was chosen as a synbiotic feed additive for large scale feeding trial on animal performance and health. 

## Figures and Tables

**Figure 1 microorganisms-07-00684-f001:**
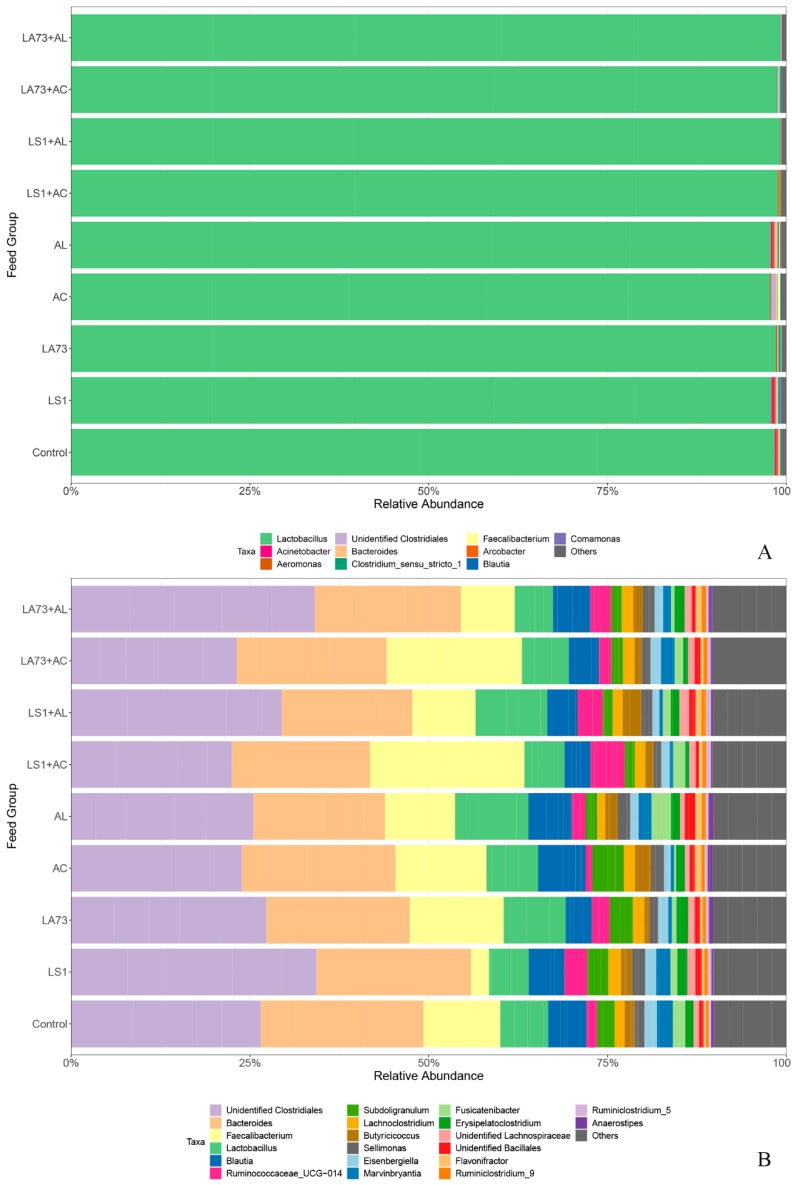
Phylogenetic composition of the intestinal microbiota in young broiler chickens. (**A**) Genus distribution in the crop; (**B**) Genus distribution in the caecum (LS1: *L.salivarius*, LA73: *L. agilis*, AC: formulation C, AL: formulation L). Data of each group are presented as the mean of five samples.

**Figure 2 microorganisms-07-00684-f002:**
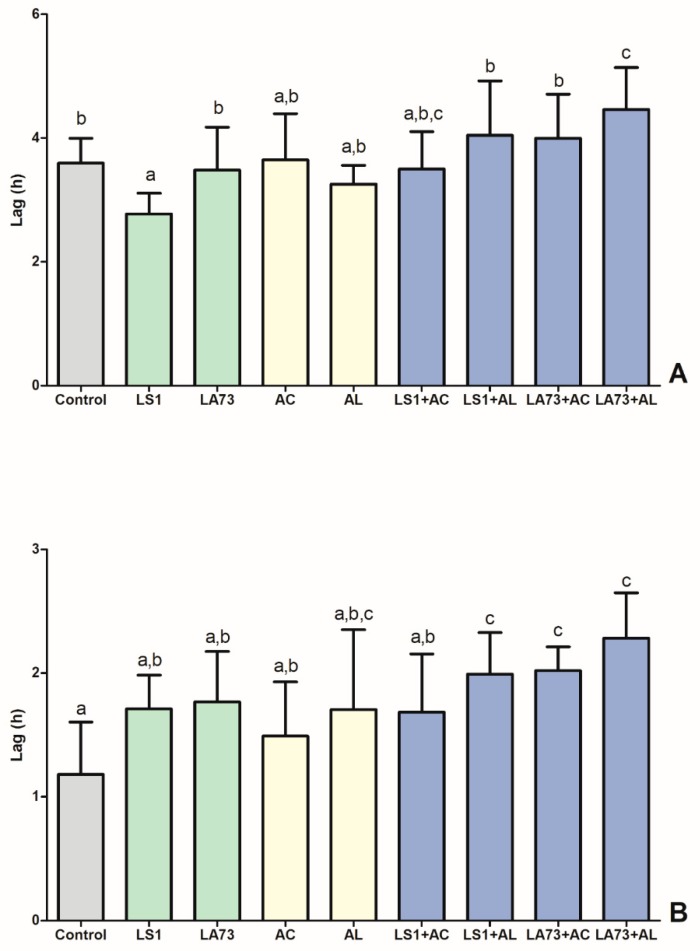
Ex vivo survival of an ESBL-producing *E. coli* strain in crop- or ceca contents of 13-day-old broiler chickens. (**A**) crop; (**B**) caecum (LS1: *L.salivarius*, LA73: *L. agilis*, AC: formulation C, AL: formulation L). Grey: Control, Green: Probiotic supplementation, Yellow: Phytobiotic supplementation, Blue: Combined feed additive supplementation. Data are presented as means with standard deviation.

**Table 1 microorganisms-07-00684-t001:** Feed ingredients and nutrition composition of the diets (as-fed basis).

Ingredient (%)	
Soybean Meal (49% CP)	32.33
Maize	32.03
Wheat	24.78
Soy oil	5.95
Limestone	1.46
Monocalcium Phosphate	1.84
Vitamin and Mineral Premix	1.20
Salt	0.10
DL-Methionine	0.18
L-Lysine	0.13
Nutrient Composition	
Crude Protein (%)	22.00
Crude Fat (%)	8.19
Crude Fiber (%)	2.42
Methionine (%)	0.51
Lysine (%)	1.28
Threonine (%)	0.84
Calcium (%)	0.96
Phosphorus (%)	0.80
AME_N_ (MJ/kg)^3^	12.6

1) Contents per kg diet: 4800 IU vit. A; 480 IU vit. D3; 96 mg vit. E (α-tocopherole acetate); 3.6 mg vit. K3; 3 mg vit. B13 mg vit. B2; 30 mg nicotinic acid; 4.8 mg vit. B6; 24 µg vit. B12; 300 µg biotin; 12 mg calcium pantothenic acid; 1.2 mg folic acid; 960 mg choline chloride; 60 mg Zn (zinc oxide); 24 mg Fe (iron carbonate); 72 mg Mn (manganese oxide); 14.4 mg Cu (copper sulfate-pentahydrate); 0.54 mg I (calcium Iodate; 0.36 mg Co (cobalt- (II)-sulfate-heptahydrate); 0.42 mg Se (sodium selenite); 1.56 g Na (sodium chloride); 0.66 g Mg (magnesium oxide). 2) Nitrogen-corrected apparent metabolizable energy estimated from the chemical composition of the feed ingredients (based on the European Union (EU) Regulation - Directive 86/174/EEC): 0.1551 × % crude protein + 0.3431 × % ether extract + 0.1669 × % starch + 0.1301 × % total sugar.

**Table 2 microorganisms-07-00684-t002:** Relative abundance of dominant putative *Lactobacillus* species in the crop of 13-day-old broiler chickens fed probiotics and phytobiotics [%].

Putative species name	Control	LS1	LA73	Formulation C	Formulation L	LS1 & Formulation C	LS1 & Formulation L	LA73 & Formulation C	LA73 & Formulation L	Pooled SEM	*p*-value^1^
*L. salivarius*	33.8^a,b^	46.5^a,b^	24.9^a,b^	68.2^b,c^	26.3^a,b^	57.1^b,c^	71.9^c^	18.1^a^	37.6^b^	3.21	0.002
*L. agilis*	16.1^a,b^	9.1^a^	38.1^b^	4.5^a^	38.3^b^	13.3^a,b^	5.7^a^	49.4^c^	20.8^a,b^	0.97	0.003
*L. crispatus*	18.2^c^	10.3^b,c^	9.9^b,c^	8.7^b,c^	5.7^a^	7.1^a^	3.2^a^	13.8^b,c^	8.4^b^	0.68	0.049
*L. reuteri*	15.3	18.0	15.2	6.2	14.0	10.9	8.2	7.2	16.5	0.61	0.425
*L. johnsonii*	8.7	9.8	8.0	9.5	11.6	9.8	8.7	5.2	11.1	1.39	0.593
*L. kitasatonis*	4.4	4.3	1.1	0.6	1.9	0.4	1.2	n.d.	0.7	3.73	0.814
*L. vaginalis*	3.4	1.9	2.9	2.4	2.1	1.5	1.1	6.3	4.7	0.43	0.120

n.d. = Not detected; ^1^ = Kruskal–Wallis Test; ^a,b^ = different superscripts denote significant changes within a row (Mann–Whitney U test, *p* ≤ 0.05).

**Table 3 microorganisms-07-00684-t003:** Quantitative determination of the crop microbiota of 13-day-old broiler chickens fed different probiotics and phytobiotics [log 16S rDNA copy number/g].

	Control	LS1	LA73	Formulation C	Formulation L	LS1 & Formulation C	LS1 & Formulation L	LA73 & Formulation C	LA73 & Formulation L	Pooled SEM	*p*- value^1^
*Lactobacillus* spp.	9.84	9.81	10.07	9.81	9.77	9.81	10.01	9.89	10.00	0.04	0.167
*L. salivarius*	9.14^a^	9.55^b^	9.39^a.b^	9.58^b.c^	9.25^a^	9.6^b.c^	9.92^c^	9.20^a^	9.53^a.b^	0.05	0.016
*L. agilis*	8.87^b^	8.83^b^	9.30^d^	8.19^a^	9.14^c^	8.73^b^	8.61^b^	9.40^d^	9.13^c^	0.11	0.016
*L. reuteri*	9.19	9.23	9.61	8.91	9.08	9.11	9.23	9.29	9.52	0.07	0.338
*L. acidophilus*	8.45	8.33	8.58	7.99	8.12	7.98	8.08	7.91	8.10	0.11	0.768
Clostridial Cluster XIVa	8.38^b^	8.41^b^	8.45^b^	8.60^b^	8.43^b^	7.76^a^	7.88^a^	8.49^b^	8.22^b^	0.09	0.040
Clostridial Cluster IV	8.08	7.89	8.11	8.16	7.86	7.36	7.43	7.92	7.72	0.09	0.206
Clostridial Cluster I	7.41^b^	7.32^b^	7.81^c^	6.95^a.b^	7.27^b^	6.99^b^	7.27^b^	7.03^b^	6.62^a^	0.08	0.016
BPP-Cluster^2^	8.48	8.36	8.56	8.22	8.37	7.95	8.16	8.29	7.82	0.07	0.338
Enterobacteria^3^	8.83^a.b^	8.42^a^	8.94^b^	8.64^a^	8.38^a^	8.30^a^	8.47^a^	8.62^a.b^	8.45^a^	0.04	0.004
*Escherichia* group	6.89^a^	6.66^a.b^	7.28^c^	6.93^a.b^	6.27^a^	6.06^a^	6.62^a^	7.01^b.c^	6.40^a^	0.08	0.018
Int1^4^	7.77^b^	7.37^b^	7.98^c^	7.27^b^	7.34^b^	7.17^b^	7.27^b^	6.99^a.b^	6.72^a^	0.08	0.042

^1^ = Kruskal–Wallis Test; ^2^ = *Bacteroides-Prevotella-Porphyromonas* Cluster; ^3^ = Copy number of the enterobacterial ribosomal polymerase beta subunit; ^4^ = Enterobacterial Integrase 1 gene.

**Table 4 microorganisms-07-00684-t004:** Quantitative determination of the caecal microbiota of 13-day-old broiler chickens fed different probiotics and phytobiotics [log 16S rDNA copy number/g].

	Control	LS1	LA73	Formulation C	Formulation L	LS1 & Formulation C	LS1 & Formulation L	LA73 & Formulation C	LA73 & Formulation L	Pooled SEM	*p*- Value^1^
Clostridial Cluster XIVa	10.96	10.97	10.85	11.15	11.09	11.08	11.17	10.93	11.11	0.05	0.639
Clostridial Cluster IV	10.39	10.28	10.51	10.72	10.61	10.73	10.67	10.47	10.30	0.05	0.134
Clostridial Cluster I	6.45	6.13	5.93	6.33	6.15	6.76	6.70	5.60	6.03	0.14	0.383
BPP-Cluster^2^	10.72	10.77	10.72	10.86	10.72	10.72	10.86	10.66	10.69	0.03	0.650
*Lactobacillus* spp.	9.63	9.61	9.82	9.73	9.83	9.69	9.94	9.61	9.67	0.04	0.605
Enterobacteria^3^	10.51	10.49	10.43	10.43	10.36	10.35	10.75	10.30	10.53	0.06	0.857
*Escherichia* group	8.45	8.43	8.32	8.21	8.53	8.13	8.13	8.20	8.43	0.07	0.910
*L. salivarius*	8.78	8.94	8.95	9.20	9.14	9.17	9.32	8.70	8.94	0.05	0.130
*L. agilis*	9.42	9.17	9.54	9.44	9.47	9.50	9.75	9.21	9.51	0.07	0.392
*L. reuteri*	8.79	8.71	9.10	8.27	8.86	8.18	8.12	9.14	9.05	0.07	0.774
*L. acidophilus*	8.26	7.48	8.21	8.12	8.49	8.27	8.27	7.67	7.79	0.11	0.620
Int1^4^	7.94	7.69	7.35	7.76	7.75	7.80	7.27	7.58	7.95	0.11	0.177

^1^ = Kruskal–Wallis Test; ^2^ = *Bacteroides-Prevotella-Porphyromonas* Cluster; ^3^ = Copy number of the enterobacterial ribosomal polymerase beta subunit.

**Table 5 microorganisms-07-00684-t005:** Concentration of lactate and short-chain fatty acid (SCFA) in the crop of 13-day-old broiler chickens fed different probiotics and phytobiotics [µmol/g].

	Control	LS1	LA73	Formulation C	Formulation L	LS1 & Formulation C	LS1 & Formulation L	LA73 & Formulation C	LA73 & Formulation L	Pooled SEM	*p*-Value^1^
L-lactate	19.8^b^	15.2^a,b^	21.1^b^	11.2^a^	14.2^a^	19.5^b^	25.0^b^	16.9^b^	29.5^c^	1.10	0.008
D-lactate	11.5^b^	3.9^a^	6.6^a,b,c^	2.2^a^	4.2 ^a^	4.9^a,b^	7.6^b^	4.8^a^	13.2^c^	0.70	0.029
total Lactate	31.3^b.c^	19.1^b^	27.8^b,c^	13.4^a^	18.3^a^	24.4^a,b^	32.6^b^	21.7^b^	42.7^c^	1.69	0.010
Acetate	5.9	4.4	6.0	2.4	3.3	4.2	4.1	4.1	6.7	0.49	0.638
Propionate	1.5	1.8	2.1	2.0	1.9	1.9	1.9	1.7	1.7	0.04	0.218
n-butyrate	0.1	n.d.^2^	0.1	n.d.	1.8	n.d.	n.d.	n.d.	n.d.	0.19	0.332
i-valerate	n.d.	n.d.	n.d.	n.d.	n.d.	n.d.	n.d.	n.d.	n.d.		0.317
n-valerate	n.d.	n.d.	0.02	n.d.	n.d.	n.d.	n.d.	n.d.	n.d.		0.277
Total SCFA	7.5	6.3	8.2	4.3	6.1	6.1	6.0	5.8	8.4	0.51	0.719
Total Metabolites^3^	40.4^b^	25.4^a,b^	35.9^b^	18.9^a^	24.5^a,b^	33.1^b^	39.5^b^	29.0^a,b^	53.7^c^	2.20	0.020

^1^ = Kruskal–Wallis Test; superscripts denote significant differences within a row (Mann–Whitney U Test, *p* ≤ 0.05); ^2^ = not detected; ^3^ = sum of the total lactate and total SCFA.

**Table 6 microorganisms-07-00684-t006:** Concentration of SCFA in the caecum of 13-day-old broiler chickens fed different probiotics and phytobiotics [µmol/g].

	Control	LS1	LA73	Formulation C	Formulation L	LS1 & Formulation C	LS1 & Formulation L	LA73 & Formulation C	LA73 & Formulation L	Pooled SEM	*p*-Value^1^
Acetate	39.9^b^	55.8^c^	32.1^a^	30.7^a^	34.8^a,b^	53.1^c^	43.6^b^	42.6^b^	44.0^b^	2.01	0.044
Propionate	5.0	7.8	6.1	5.2	4.2	5.7	5.2	5.9	7.0	0.28	0.142
i-butyrate	1.9	2.1	0.8	0.6	1.2	1.1	0.4	0.5	0.6	0.22	0.963
n-butyrate	8.7	7.7	5.8	7.7	6.0	9.4	8.7	10.1	6.2	0.50	0.548
i-valerate	0.5	0.3	0.3	0.4	0.2	0.4	0.2	0.3	0.2	0.03	0.684
n-valerate	0.4	0.4	0.6	0.6	0.3	0.4	0.3	0.2	0.3	0.04	0.504
BCFA^2^	2.0	2.4	1.1	0.8	1.2	1.5	0.6	0.6	0.7	0.21	0.873
Total SCFA	55.9	74.1	45.6	44.9	46.4	70.1	58.4	59.3	58.2	2.50	0.103

^1^ = Kruskal–Wallis Test; superscripts denote significant differences within a row (Mann–Whitney U Test, *p* ≤ 0.05).; ^2^ = Sum of branched chain fatty acids.

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
