# Peer review of "Synergistic Effects of Probiotics and Phytobiotics on the Intestinal Microbiota in Young Broiler Chicken"

_microorganisms, 2019, doi:10.3390/microorganisms7120684_

Round 1

Reviewer 1 Report

The manuscript entitled “Synergistic effects of probiotics and phytobiotics on the intestinal microbiota in young broiler chickendescribes a chicken feed intervention trial using probiotics and phytobiotics (combination) with the objective of modulating the chicken gut microflora to improve gut health and productivity.

The authors used 16S rDNA sequencing to analyse the predominant bacterial genera present in crop and caecum. To analyse at species level qPCR was used. All the qPCR data is located in supplementary results.

The metabolic activity was determined by analysing short chain fatty acid profile using gas chromatography.

To determine the efficacy of the effect of modulating the microflora using intervention strategy, extended spectrum beta lactamase (ESBL) producing E.coli growth was measured ex vivo in crop and caecum contents.

I have several serious concerns regarding the quality of this work.  These include-

The authors used so called 'host-specific Lactobacillus strains (L. agilis and L. salivarius) in their studies but provided no evidence that these specific strains have been proven to be probiotics. The authors need to know that the probiotic effects are strain specific. Thus, failure to detect any beneficial effects is not a surprise.

The 16S rDNA sequencing method or the corresponding graph legend does not mention about the number of samples used for each data point. The qPCR method/corresponding table/graph also does not provide any detail on the number of replicates tested (please check).

The method used for the growth of ESBL E coli in the crop and cecum content appear to be flawed. The diluted crop/caecal samples are sedimented (not clear if this is centrifugation), does this separate all the bacteria which means the ESBL E coli is grown in the absence of microflora. If the bacteria were not removed during sedimentation, then there is initial OD due to bacteria which could interfere with the interpretation of the lag time. Secondly, the growth of this organisms is measured in the presence of amoxycillin. This means the growth of the ESBL E coli was measured when most of the microflora of the samples are not metabolically active or dead. This does not really measure the true inhibition of the ESBL E coli in the crop/caecal content. In addition, measuring the lag phase has also not provided convincing results (growth in small volume in microtiter plate may not represent true growth).

Results and discussion:

The major genera in crop irrespective of treatment was predominantly Lactobacillus. The other genera Aeromonas and Acinetobacter compose of less than 5% of the predominant genera sequences. The authors discuss the treatment effect (line 215 to 220) with undue emphasis.

Line 342-343, the authors state that ‘L agilis strain was strongly inhibited by formulation C in vivo’ . Since gut content is complex mixture of living and non-living matter, there could be some other metabolite (produced by another group of microorganisms) that could inhibit the L agilis strain. It does not make sense to say that the formulation C has inhibited the growth ‘in vivo’ knowing that the formulation was stable in vitro (Line 340).

Line 364-365 “Significant positive synergistic effects on relative Lactobacillus spp. abundance were only observed for L. salivarius in combination with formulation L as well as for L agilis with formulation C”. This statement is incorrect since Table 3 shows that both L salivarius and L agilis growth were promoted by formulation L.

Conclusion: The authors conclude that the beneficial effect of the probiotics and phytobiotics is primarily in crop together with effect on Clostridium perfringens and reduced ex vivo survival of ESBL-producing E coli in caecum. Interestingly, the authors choose L agilis strain in combination with formulation L  for future feeding trials. Table 4 shows that in caecum, L agilis strain with formulation C inhibited the Clostridium group I to a significant extent (and more effective compared to formulation L). This inhibition was slightly greater than the inhibition found in strain L agilis with formulation L in crop.

The authors used the less convincing ESBL producing E coli inhibition test and based part of their conclusion on these results. Secondly, the authors used qPCR results (species specific data) in deriving conclusion on C perfringens and yet the data is in supplementary tables. The 16S rDNA sequencing results (genus specific sequence abundance) have not provided good resolution of the microflora in crop. The authors have generated a large quantity of data and tried to discuss these results. This has led to a lack of focus in the manuscript and created more confusion than clarity. 

Author Response

Dear reviewers

We would like to thank you for the thoughtful comments and constructive suggestions, which help to improve the quality of this manuscript. We have carefully reviewed the comments of reviewers and incorporated the necessary changes in accordance with comments in a way of point-by-point. We hope this current version would meet the publication criteria. Please refer to our response as follows

Q1

The authors used so called 'host-specific Lactobacillus strains (L. agilis and L. salivarius) in their studies but provided no evidence that these specific strains have been proven to be probiotics. The authors need to know that the probiotic effects are strain specific. Thus, failure to detect any beneficial effects is not a surprise.

Answer:  Thanks for the comments. The strains in this manuscript were selected due to their promising in vitro and ex vivo probiotic functionalities in our previous study. Thus, they were deemed as the most interesting strains for further in vivo studies. To make this clear, the study on characterization and selection of these specific strains are now mentioned in line 124, 390 and 444 to clarify the rationale for selecting these strains. We also revised the expression from line 47 to 54 as ‘Certain Lactobacillus agilis strains are able to modify the presence of pathogenic bacteria in vitro [10] and ex vivo [11] or regulate the gut microbiota in broiler chicken in vivo [12,13]. In some studies several L. salivarius strains have also been shown to promote the animal health [14,15]. Proper supplementation of certain probiotic strains may also lead to immunomodulation of the host [16] and consequently, resilience against pathogenic bacteria may be increased [17,18]. However, those effects primarily depend on the specific strain and it is still unclear, if immunomodulation is a direct effect of the probiotics or a response to a modified microbiota’. This emphasizes the notion that the effect primarily depend on specific strains and must be viewed individually.

Q2

The 16S rDNA sequencing method or the corresponding graph legend does not mention about the number of samples used for each data point. The qPCR method/corresponding table/graph also does not provide any detail on the number of replicates tested (please check).

Answer: Thanks for the comments. The details regarding the sample scale and replicate number are now added to convince the validation of data in the method section. We revised the statement in the method section in line 146-147 as ‘Total DNA was extracted from 0.2 g digesta samples of crop and caecum (45 samples for each section, 5 samples for each group) with a commercial extraction kit’. We also changed the caption of Figure 1 in line 212 to read: ‘Data of each group is presented as mean of five samples.’ Regarding the information of the qPCR procedure, we added a supplementary statement in line 160-161 as ‘Quantification of predominant bacterial groups and species in poultry was performed via qPCR assays with five replicates per group’.

Q3

The method used for the growth of ESBL E coli in the crop and cecum content appear to be flawed. The diluted crop/caecal samples are sedimented (not clear if this is centrifugation), does this separate all the bacteria which means the ESBL E coli is grown in the absence of microflora. If the bacteria were not removed during sedimentation, then there is initial OD due to bacteria which could interfere with the interpretation of the lag time. Secondly, the growth of this organisms is measured in the presence of amoxycillin. This means the growth of the ESBL E coli was measured when most of the microflora of the samples are not metabolically active or dead. This does not really measure the true inhibition of the ESBL E coli in the crop/caecal content. In addition, measuring the lag phase has also not provided convincing results (growth in small volume in microtiter plate may not represent true growth).

Answer: Thanks for the comments. We have considered these questions when we established the current method, which is based on a validated and published method. To answer the comments:

Sedimentation refers to the process in which particles separate from liquid via gravity (centrifugation at 1g). In this case, this has been done to separate large feed particles from the liquid phase, which would interfere with OD measurements. Sedimentation time was validated in the mentioned study as pre-experiments (data are not shown in that study). During that validation, an acceptable sedimentation period after homogenizing samples was found to be 5 minutes. Sedimentation (1g) is not able to separate bacteria in the liquid phase and is thus a valid method to separate large particles from the microbiota. Of course, bacteria that are strongly attached to feed particles will be lost, but the amount of bacteria that are strongly attached to feed particles is usually low in monogastric animals, this is different for some ruminal bacteria, for instance. Therefore, the majority of bacteria is still present in the supernatant and should be considered as the main microbiota that interacts with inoculated bacteria (here coli). They were not designed to be excluded but kept as important ex vivo factor. Incidentally, this was also discussed with a reviewer during the revision of the mentioned published study. The time-dependent OD due to the presence of indigenous microorganisms as well as potentially-existing ESBL bacteria was corrected by the use of appropriate controls. In the last version, we didn’t describe the details of the setup of control plates which were done in the experiment. To clarify this, we added the following (line 176-178): “To avoid interference from indigenous bacteria in the samples, identical plates but without inoculation of the ESBL indicator strain was prepared and incubated under the same conditions.” and (line 183): “was obtained as OD experiment subtracted by OD control’. Moreover, any initial OD interference was rendered irrelevant since they were quite low to OD690nm=0.08-0.15 due to the high dilution. The ex vivo experiment comprises with two consecutive incubations. The aim of adding cefotaxime during the second step of the incubation (the first incubation step does not contain antibiotics) was to selectively allow growth of the inoculated ESBL coli to monitor its fitness after incubation under existing ex-vivo conditions (the first incubation step) and to record its inhibitory capacity. The first co-incubation was totally antibiotic-free and the organisms in ex vivo model wouldn’t be influenced. To clarify this, we added the term “in a two-step incubation assay” and “in a first step” to line 173 as well as “In the second step” in line 178-179. The absence of growth in control samples (incubation without the E. coli strain) shows that indeed only the inoculated E. coli strain was able to grow in the cefotaxime supplemented medium. Regarding the obtained data on lag increase: We are of course aware of the shortcomings of (any) in vitro However, lag time is a very clear-cut indicator of the physiological state of an initial cell population, i.e. its fitness. This is why exponentially growing cultures exhibit almost no lag time at all upon re-inoculation, but cultures from, for instance, cryo cultures display quite large lag times. Furthermore, cell density also defines lag time and thus, taken together with initial physiological state infer a parameter on general fitness of the inoculum. This was what we were interested in: how much does a pre-incubation in different sample matrices reduce the fitness of the inoculated strain? The employed ex vivo method also has technical merits such as good practice of animal welfare, low on hand time and more importantly high replicates numbers. The similar ex vivo studies have been employed with diluted slurry or bioreactor in several studies [1-4]. Hence, we believe that it can be regarded as a useful tool to study microbiota interaction and bacterial fitness in our current study. However, a detailed discussion of this method was beyond the scope of this manuscript and we therefore referred the reader to our previous study. Starke, I.C.; Zentek, J.; Vahjen, W. Ex vivo-growth response of porcine small intestinal bacterial communities to pharmacological doses of dietary zinc oxide. PloS one 2013, 8, e56405-e56405, doi:10.1371/journal.pone.0056405. Hao, R.; Eva-Maria, S.; Jürgen, Z.; Farshad, G.B.; Wilfried, V. Screening of host specific lactic acid bacteria active against Escherichia coli from massive sample pools with a combination of in vitro and ex vivo methods (in press). Front Microbiol 2019, 10: 2705, doi:10.3389/fmicb.2019.02705. Ghadimi, D.; Ebsen, M.; Kabisch, J.; Röcken, C.; Moghadasian, M.H.; Heller, K.J.; Bockelmann, W.; Franz, C. In vitro, ex vivo and in vivo effects of egg consumption (and egg microbiota) on the microbial community composition of gut microbiota and on immunological/metabolic/inflammatory biomarkers. Z Gastroenterol 2015, 53, G6, doi:10.1055/s-0035-1558904. Reese, A.T.; Cho, E.H.; Klitzman, B.; Nichols, S.P.; Wisniewski, N.A.; Villa, M.M.; Durand, H.K.; Jiang, S.; Midani, F.S.; Nimmagadda, S.N., et al. Antibiotic-induced changes in the microbiota disrupt redox dynamics in the gut. eLife 2018, 7, e35987, doi:10.7554/eLife.35987.

Q4

The major genera in crop irrespective of treatment was predominantly Lactobacillus. The other genera Aeromonas and Acinetobacter compose of less than 5% of the predominant genera sequences. The authors discuss the treatment effect (line 215 to 220) with undue emphasis.

Answer: Thanks for the comments. Due to the nature of the crop, the microbiota composition is largely predominant by the Lactobacillus genus. This was also observed several other studies and is a commonly known fact that we did not discuss in detail [5]. However, most studies only analyzed at the genus level. We therefore distinguished unique Lactobacillus OTUs to further our knowledge of dominating lactobacilli at the OTU level. We did find that the added lactobacilli specifically and significantly increased their respective abundances. However, we agree that there might be some redundancy and our presentation of the data could be more concise. Thus, we removed the overdue expression in line 233-236 and 239-241. We revised the sentence in line 236 as ‘Seven different Lactobacillus species were assigned based on their unique OTUs…’ and moved the rest to line 222-226.

Shang, Y.; Kumar, S.; Oakley, B.; Kim, W.K. Chicken Gut Microbiota: Importance and Detection Technology. Front Vet Sci 2018, 5, 254-254, doi:10.3389/fvets.2018.00254.

Q5

Line 342-343, the authors state that ‘L agilis strain was strongly inhibited by formulation C in vivo’ . Since gut content is complex mixture of living and non-living matter, there could be some other metabolite (produced by another group of microorganisms) that could inhibit the L agilis strain. It does not make sense to say that the formulation C has inhibited the growth ‘in vivo’ knowing that the formulation was stable in vitro (Line 340).

Answer: Thanks for the comments. We agree that our expression regarding the L. agilis inhibition is not precise, as not exactly the formulation C itself, but the effect of the formulation led to the inhibition. However, we find that the single application of formulation C drastically decreased L. agilis in the crop while the combination of formulation C with L. agilis increases L. agilis. This suggests that indigenous L. agilis might be more sensitive to the formulation C but the supplemented L. agilis is obviously stable against this formulation. This also indicates that strain-specific properties of lactobacilli exist. Based on your comments, we revised this argument as follows (lines 362 to 369): ‘ However, the species L. agilis was strongly inhibited be effects of the formulation C in vivo, but in combination with LA73 an increase of this species was observed  in the crop. This also points to a synergistic effect for an increased LA73 colonization in combination with formulation C. Taken together, the data suggests that indigenous L. agilis strains may be much more sensitive to phytobiotic pressure compared to the supplemented L. agilis strain. This reflects strain-specific differences of lactobacilli in general. Therefore, synergistic effects seem to be in effect regarding certain combinations of probiotic and phytobiotics.’

Q6

Line 364-365 “Significant positive synergistic effects on relative Lactobacillus spp. abundance were only observed for L. salivarius in combination with formulation L as well as for L agilis with formulation C”. This statement is incorrect since Table 3 shows that both L salivarius and L agilis growth were promoted by formulation L.

Answer: We thank you for your observation and revised this accordingly (line 393): ‘both phytobiotics’.

Q7 

Conclusion: The authors conclude that the beneficial effect of the probiotics and phytobiotics is primarily in crop together with effect on Clostridium perfringens and reduced ex vivo survival of ESBL-producing E coli in caecum. Interestingly, the authors choose L agilis strain in combination with formulation L for future feeding trials. Table 4 shows that in caecum, L agilis strain with formulation C inhibited the Clostridium group I to a significant extent (and more effective compared to formulation L). This inhibition was slightly greater than the inhibition found in strain L agilis with formulation L in crop.

Answer: Thanks for the comments. We based our conclusion on the use of L. agilis and formulation L on the evaluation of all available data. The Clostridial Cluster I is one of the predominant groups and includes many different species. There are pathogenic members like Clostridium botulinum, but also members like Clostridium butyricum which used to be reported as both probiotic or pathogens [6,7]. Thus, we cannot clearly assign this cluster as a beneficial or detrimental. However, we also noticed our conclusions are not clear enough and thus revised it as follows (line 492 to 496): ‘Comprehensively considering the effects in microbiota shifts, bacterial metabolite changes and resilience to detrimental bacteria in host GIT, the combination of the L. agilis strain in combination with the formulation L was chosen as a synbiotic feed additive for large scale feeding trial on animal performance and health’.

Dohrmann, A.B.; Walz, M.; Löwen, A.; Tebbe, C.C. Clostridium cluster I and their pathogenic members in a full-scale operating biogas plant. Applied Microbiology and Biotechnology 2015, 99, 3585-3598, doi:10.1007/s00253-014-6261-y. Cassir, N.; Benamar, S.; La Scola, B. Clostridium butyricum: from beneficial to a new emerging pathogen. Microbiol. Infect. 2016, 22, 37-45, doi:https://doi.org/10.1016/j.cmi.2015.10.014.

Q8

The authors used the less convincing ESBL producing E coli inhibition test and based part of their conclusion on these results. Secondly, the authors used qPCR results (species specific data) in deriving conclusion on C perfringens and yet the data is in supplementary tables. The 16S rDNA sequencing results (genus specific sequence abundance) have not provided good resolution of the microflora in crop. The authors have generated a large quantity of data and tried to discuss these results. This has led to a lack of focus in the manuscript and created more confusion than clarity. 

Answer: Thanks for the comments. We explained and revised the topic on E. coli inhibition with an ex vivo model in answers for Q3. With this necessary revision, we believe it is now clear that the well-established method which was used in many previous studies is appropriate to study inhibition of E. coli. We placed the results on C. perfringens into supplementary information, as it was a follow up on the relative abundance of Clostridium sensu stricto 1, which is phylogenetically close to C. perfringens. Therefore, it has been deemed secondary to other findings.

As to the sequencing results on the crop microbiota, this study is actually one of the first study that tried to analyze the dominant lactobacilli on the OTU level. Due to the absolute dominance of lactobacilli in the crop, a deeper resolution than in our study can only be achieved with true metagenomic analysis with massive sequence depth, which is prohibitely high in costs for such a large sample number and was not the prime parameter of this study. Therefore, we believe that by analyzing the OTU level of lactobacilli is a good balance between resolution and aim of the study.

We carefully revised the expression and data presentation throughout the manuscript. In this revision and based on comments of the other reviewers we believe that the manuscript is batter explained and now displays a higher level of focus and clarity.

Reviewer 2 Report

Authors Ren et al. submitted their manuscript entitled “Synergistic effects of probiotics and phytobiotics on the intestinal microbiota in young broiler chicken”. The modulation of a gastrointestinal microbiota is an actual and interesting topic, if bacterial resistance is increased, using of antibiotics as feeding additives are banned in some countries, and using of other antibacterial compounds as ZnO will be banned. It is necessary to find and use other ways to keep the health of humans and animals. The using of probiotics, phytobiotics, and their combinations belong to promising ways of future research.  

The manuscript deals with using two lactobacilli with probiotic properties, two phytobiotic preparations, and their combination in young broiler chicken. The topic is actual and interesting, the manuscript is carefully prepared, and well written.

L26: „... bacterial composition and activity...". Please, clarify which activity do you mean.

L48-50: It is known that different strains of lactobacilli can show different effects on the host. I am not sure if it is possible to generalize lactobacilli species that caused such an effect.

L121 - I think that it would be better - at first to characterize lactobacilli (in1st paper) and at second (in 2nd paper) to use characterized ones. You reversed it.

L131-132: Please, precise if you cultured bacteria on agar or broth, washed a pellet ...

L132: 15,000 g, 10 min seems to be high speed for a long time. Please, verify it.

L182-182: You wrote that the results are presented as means ± standard deviation SD or SEM. Statistical significance of normally distributed data was assessed using one-way ANOVA)and Tukey test. Statistical significance of non-normally distributed data was analyzed with the Kruskal-Wallis test, followed by the Mann-Whitney test.

i) The results are not presented in all cases as means ± SD or SEM. E.g., they are presented as mean + SD or SEM in a column graph in Figure 4. ii) Mean, SD, and SEM are characteristics of parametric tests but not non-parametric ones. Please, distinguish it.

L202, Figure 1: It is a little confusing that the same color labels the most abundant bacteria. It would be better to use the same colors for the same bacterial groups to be clear that the microbiota composition in the crop and cecum are different.

L267-268: Please, rephrase the sentence. If it was significant, "slightly" is inappropriate.

L301: It is not clear which is the difference between "significantly" and "numerically". If numerically means non-significantly, it should be written.

L324, Figure 4: It is not clear what is depicted in this column graph. You wrote that you use it. I notice again that mean, SD, and SEM are characteristics of parametric tests but not non-parametric ones (median, range, etc.). Please, describe in the figure legend what is depicted in Figure 4.

L363: A basic characterization of both used lactobacilli should be suitable.

L396-397: The results differ statistically significantly, or they did not differ. What does it mean "numerically". If it means statistically non-significantly, please, remove numerically.

L414-415: This manuscript dealing with inhibitory properties of the used lactobacilli to the E. coli should come before this submitted manuscript.

Author Response

Dear reviewer

We would like to thank you all for the thoughtful comments and constructive suggestions, which help to improve the quality of this manuscript. We have carefully reviewed the comments of reviewers and incorporated the necessary changes in accordance with comments in a way of point-by-point. We hope this current version would meet the publication criteria. Please refer to our response as follows.

Comments and Suggestions for Authors

Authors Ren et al. submitted their manuscript entitled “Synergistic effects of probiotics and phytobiotics on the intestinal microbiota in young broiler chicken”. The modulation of a gastrointestinal microbiota is an actual and interesting topic, if bacterial resistance is increased, using of antibiotics as feeding additives are banned in some countries, and using of other antibacterial compounds as ZnO will be banned. It is necessary to find and use other ways to keep the health of humans and animals. The using of probiotics, phytobiotics, and their combinations belong to promising ways of future research.  

The manuscript deals with using two lactobacilli with probiotic properties, two phytobiotic preparations, and their combination in young broiler chicken. The topic is actual and interesting, the manuscript is carefully prepared, and well written.

Q1

L26: „... bacterial composition and activity...". Please, clarify which activity do you mean.

Answer: Thanks for the comments, we have revised the expression in line 26 as ‘bacterial composition and their metabolic activity’.

Q2

L48-50: It is known that different strains of lactobacilli can show different effects on the host. I am not sure if it is possible to generalize lactobacilli species that caused such an effect.

Answer: Thanks for this comments, we agree that the effect of lactobacilli species cannot be generalized but depend on strain-specific behavior. Therefore, we have revised the sentences from line 47 to 51 as ‘Certain Lactobacillus agilis strains are able to modify the presence of pathogenic bacteria in vitro [10] and ex vivo [11] or regulate the gut microbiota in broiler chicken in vivo [12,13]. In some studies several L. salivarius strains have also been shown to promote the animal health [14,15]. Proper supplementation of certain probiotic strains may also lead to immunomodulation of the host …’

Q3

L121 - I think that it would be better - at first to characterize lactobacilli (in1st paper) and at second (in 2nd paper) to use characterized ones. You reversed it.

Answer: Thanks for the comments and we apologize for the confusion. Both probiotic strains were selected in our previous work. The selection and characterization of both strains were summarized and submitted previously and are in press now (doi: 10.3389/fmicb.2019.02705). We cite this report to clarify that the selection progress and probiotic properties of the both strains have been published.

Q4

L131-132: Please, precise if you cultured bacteria on agar or broth, washed a pellet ...

Answer: Thanks for the comments. The cultivation of the both probiotic strains was done in liquid broth medium. The biomass (pellets) was then concentrated by centrifugation. To make this clear, we made minor modification in line 132-133 as ‘The probiotic Lactobacillus cells were harvested from MRS broth after aerobic growth at 37 ºC for 12 h. The pelleted biomass was concentrated by centrifugation …’. To maintain the manuscript concise, the description of production of probiotic strains as feed additives were not given in detail here. We referred to our previous publication in line 135, which provided the in-depth information about how we prepared and formulated the probiotic strains.

Q5

L132: 15,000 g, 10 min seems to be high speed for a long time. Please, verify it.

Answer: Thanks for the comments. Different centrifugation procedures of lactobacilli strains were reported in different papers. The determination of the centrifugation was performed in our past works, we mainly referred to two published reports (see below) and found that the 15,000 g for 10 min at 4 °C is more suitable for concentration of lactobacilli pellets on a large scale to increase cell yield. Therefore, this procedure has been determined and used in our lab as standardized procedure to produce the lactobacilli pellets.

References

Garcia, E.F., Luciano, W.A., Xavier, D.E., da Costa, W.C.A., de Sousa Oliveira, K., Franco, O.L., et al. (2016). Identification of Lactic Acid Bacteria in Fruit Pulp Processing Byproducts and Potential Probiotic Properties of Selected Lactobacillus Strains. Frontiers in Microbiology 7(1371). doi: 10.3389/fmicb.2016.01371.

Shehata, M.G., El Sohaimy, S.A., El-Sahn, M.A., and Youssef, M.M. (2016). Screening of isolated potential probiotic lactic acid bacteria for cholesterol lowering property and bile salt hydrolase activity. Annals of Agricultural Sciences 61(1), 65-75. doi: https://doi.org/10.1016/j.aoas.2016.03.001.

Q6

L182-182: You wrote that the results are presented as means ± standard deviation SD or SEM. Statistical significance of normally distributed data was assessed using one-way ANOVA)and Tukey test. Statistical significance of non-normally distributed data was analyzed with the Kruskal-Wallis test, followed by the Mann-Whitney test.

i) The results are not presented in all cases as means ± SD or SEM. E.g., they are presented as mean + SD or SEM in a column graph in Figure 4. ii) Mean, SD, and SEM are characteristics of parametric tests but not non-parametric ones. Please, distinguish it.

Answer:

i) We apologize for the misunderstanding. We omitted standard deviations from tables, as SD makes the tables hard to read. We used SD only in figure 2 and have added the following to the legend (L331-332): “Data is presented as means and standard deviation.” ii) We agree that means, SD or SEM are used in parametric tests. We chose to present means and SEM for clarity, but as most data was not normally distributed, we used the more robust non-parametric Kruskal-Wallis/ Mann-Whitney tests. We apologize for the ‘copy&paste error” and have therefore rewritten the statistics as follows (Line 184-192): “Results are presented as means and pooled standard error of mean except for figure 2, which is presented as means and standard deviation. Due to the non-normally distributed nature of the data we chose to use Kruskal-Wallis test, followed by the Mann-Whitney test, when appropriate. Chi-square test was performed to compare C. perfringens counts. Statistical procedures were performed at a significance level of 95% using SPSS Statistics software (SPSS, Chicago, USA).’

Q7

L202, Figure 1: It is a little confusing that the same color labels the most abundant bacteria. It would be better to use the same colors for the same bacterial groups to be clear that the microbiota composition in the crop and cecum are different.

Answer: Thanks for the comments. To avoid confusion caused by inconsistent color assignment, we reproduced the bar plots to unify the color labels.

Q8

L267-268: Please, rephrase the sentence. If it was significant, "slightly" is inappropriate.

Answer: Thanks for the comments, we agree that the ‘slight’ here may introduce confusing expression followed with expression ‘significantly’. Therefore, we rephrase the sentence in 271-272 as ‘Copy numbers of total enterobacteria and the Escherichia group were higher in the LA73 group, but absolute differences were only minor.’.

Q9

L301: It is not clear which is the difference between "significantly" and "numerically". If numerically means non-significantly, it should be written.

Answer: Thanks for the comments. The formulation C decreased the acetate significantly, yet the reduction induced by formulation L was only numeric. To make this statement clear, we revised ‘numerically’ as ‘numerically but non-significantly’ in line 305.

Q10

L324, Figure 4: It is not clear what is depicted in this column graph. You wrote that you use it. I notice again that mean, SD, and SEM are characteristics of parametric tests but not non-parametric ones (median, range, etc.). Please, describe in the figure legend what is depicted in Figure 4.

Answer: Thanks for the comments. We apologize for the confusing caused by the deficiency in figure captions. We used SD for the columns in figure 2 (both A and B) and have added the following to the legend (line331-332): “Data is presented as means and standard deviation.” As we explained in answers for Q6, we use non-parametric Kruskal-Wallis/ Mann-Whitney tests for the statistical analysis as data were not normally distributed. Additionally, we also added the captions to clarify the color labels which reflects the feed types in lines 330-331.

Q11

L363: A basic characterization of both used lactobacilli should be suitable.

Answer: Thanks for the comments. The sentence alone here seems not strong enough to support the conclusion here. We picked up your suggestions to cite their characterization showing great potential to survive under the gastric stress and adhere the epithelia in in vitro condition. We revised the expression from lines 380 to 382 as ‘This was not unexpected, as both probiotic strains were originally isolated from the broiler intestine and already demonstrated great potential for in vitro survival under stimulated gastric stress and epithelial adherence in our previous study’, we also added our article in press as the reference here.

Q12

L396-397: The results differ statistically significantly, or they did not differ. What does it mean "numerically". If it means statistically non-significantly, please, remove numerically.

Answer: Yes, we agree the usage of ‘numerically’ to address the non-significant changes may not be precise. We removed the term of ‘numerically’ and rephrase the sentence in lines 417 to 419 as ‘Overall, bacterial activity in terms of bacterial metabolites was lower in the crop of feed groups with single phytobiotic addition, although only the reduction of lactate was significant.’ Additionally, the term ‘numerical’ in line 387 was also revised as ‘non-significant’.

Q13

L414-415: This manuscript dealing with inhibitory properties of the used lactobacilli to the E. coli should come before this submitted manuscript.

Answer: Thanks for the suggestion. The reviewing progress of our previous work was longer than expected, but the manuscript is in press now. The mentioned antagonistic properties of used lactobacilli strains in in vitro and ex vivo conditions are referred in line 436. The previous status of ‘submitted’ is now removed and replaced with reference information provided by the publisher.

Reviewer 3 Report

The outcome of the article is very good. The analytical part is sufficient. 

However I have a question to the authors.

Is the outcome much better than the one comes from the single application of microorganisms?

Specifically, the authors write” Lactobacillus agilis strains were reported to modify the presence of pathogenic 48 bacteria in vitro [10] and ex vivo [11] or regulate the gut microbiota in broiler chicken in vivo [12,13]."

Similar results have been reported for L. salivarius strains [14,15].” Likewise the two microorganisms are functional in single from

I believe that the authors should present more clear the outcome regarding this issue, which is of great importance related to the level of originallity and economic feasibility of the whole proposed methology

Author Response

Dear reviewer

Thank you very much for taking your time to review this manuscript. I really appreciate all your comments and suggestions. Please find my itemized responses in below and my revisions in are-submitted manuscript.

Q1

The outcome of the article is very good. The analytical part is sufficient. 

However I have a question to the authors.

Is the outcome much better than the one comes from the single application of microorganisms?

Specifically, the authors write” Lactobacillus agilis strains were reported to modify the presence of pathogenic 48 bacteria in vitro [10] and ex vivo [11] or regulate the gut microbiota in broiler chicken in vivo [12,13]."

Similar results have been reported for L. salivarius strains [14,15].” Likewise the two microorganisms are functional in single from

I believe that the authors should present more clear the outcome regarding this issue, which is of great importance related to the level of originallity and economic feasibility of the whole proposed methology

Answer: Thanks for the comments. We have tried to incorporate this issues in a more concise manner. As we explained in the introduction (line 65 to 71), there are quite limited studies addressing a possible synergistic mode of action with probiotics and phytobiotics in broiler chickens. Therefore, we highlighted this issue (lines 73 to 75): ‘Therefore, a concept was designed and applied in this study to combine host-specific probiotic Lactobacillus strains with specific phytobiotics to invoke beneficial synergistic effects for the animal.’ We also changed some expression in the introduction for clarity of strain-specific manner of probiotic strains (lines 47-51).

To better clarify the rationale why emphasizing the synergic application, we added the statement from line 334 to 338 as ‘Accumulating numbers of studies support the alternative role of novel additives such as probiotics or phytobiotics to the in-feed AGPs. However, the efficiency of the alternative additives depend on many factors like uptake concentration, overall diet, supplementation method or rearing environment [50]. To maximize the efficiency of those alternatives, the combination in accordance with synergistic concept is a favorable solution may act beyond their single applications.’

Concerning the discussion, we rephrased and clarified the argument that there seem to be synergistic effects for certain strains and certain formulations regarding their colonization (line 352 to 361): ‘ In fact, a strong synergistic effect was observed for the species L. salivarius with formulation L, which may indicate that the functionality of LS1 increased accordingly, when applied as a synergistic product. However, the species L. agilis was strongly inhibited by formulation C in vivo, but the combination of LA73 and formulation C increased the amount of this species in the crop and caecum. This also points to a synergistic effect for an increased LA73 colonization in combination with formulation C. Taken together, the data suggests that the colonization of the probiotics and their resilience against the phytobiotics cannot be deduced from in vitro studies alone. Furthermore, there is a strong indication that indigenous species are reduced in favor of the supplemented probiotic strains. Therefore, synergistic effects seem to be in effect regarding certain combinations of probiotic and phytobiotics.‘

We also included a conclusion on bacterial composition to highlight the possible beneficial outcome of probiotic and phytobiotics (line 414 to 416):” However, as beneficial synergistic effects are clearly visible regarding putatively pathogenic bacteria, the combination of certain pro- and phytobiotic products may be advantageous for animal health.“ and (line 449 to 451): “ Nevertheless, the results clearly show that synergistic effects of probiotics and phytobiotics may be superior to single addition to combat the spread of enterobacterial antibiotic resistance.“

By comparing the feed additives as combination with single application, we found that certain combination of probiotic with phytobiotic (LA73 with formulation L in the present study) revealed the enhanced capability to regulate the GIT microbiome, improve the bacterial metabolic activity (lactate), reduce the fitness of exogenous pathogens (ESBL E. coli or C. pfringens) in host GIT ecology and decreased the expression of important enzyme that mediate resistance transmission (Int1) than control and single applications. The results bring the insights of positive role of synergic combination as feed additive in our case, subsequently endorsed the potential mode of probiotics-phytobioics combination in health promotion. To briefly highlight the superior performance of combinations over single application and control, we revised statement in 486 to 487 as ‘…chosen as a synbiotic feed additive for large scale feeding trial on animal performance and health.’

We hope that these modifications better explained the rationale behind the focus on synbiotic combination of probiotics and phytobiotic products and clarify the outcomes in accordance with purpose.

Round 2

Reviewer 3 Report

The authors corrected the sproposed suggestions. Therefore I believe that the article should be accepted

Author Response

Dear reviewer

We thank you for your kind efforts and time on the current manuscript.

Best regards,

All authors